# Accurate Realtime Motion Estimation Using Optical Flow on an Embedded System

**Anis Ammar** [1,*] , **Hana Ben Fredj** [2] **and Chokri Souani** [3]

1   Laboratoire de Microelectronique et Instrumentation, Ecole National d'Ingineurs de Sousse,
   Université de Sousse, Sousse 4054, Tunisia
2   Laboratoire de Microelectronique et Instrumentation, Faculté de Sciences de Monastir, Université of Monastir,
   Monastir 1002, Tunisia; ben.fredj.hanaa@gmail.com
3   Institut Supérieur des Sciences Appliquées et Technologiques de Sousse, Université of Sousse,
   Sousse 4003, Tunisia; chokri.souani@gmail.com
*   Correspondence: anis.ammarr@gmail.com

**Abstract:** Motion estimation has become one of the most important techniques used in realtime computer vision application. There are several algorithms to estimate object motions. One of the most widespread techniques consists of calculating the apparent velocity field observed between two successive images of the same scene, known as the optical flow. However, the high accuracy of dense optical flow estimation is costly in run time. In this context, we designed an accurate motion estimation system based on the calculation of the optical flow of a moving object using the Lucas–Kanade algorithm. Our approach was applied on a local treatment region implemented into Raspberry Pi 4, with several improvements. The efficiency of our accurate realtime implementation was demonstrated by the experimental results, showing better performance than with the conventional calculation.

**Keywords:** motion estimation; optical flow; embedded system; realtime implementation





## 1. Introduction

Motion estimation is an important task in the basic research field of automated video surveillance systems in computer vision and video processing. To make these systems more effective, they should respond to realtime constraints with a high performance [1]. Therefore, realtime embedded systems in scientific research try to find the best compromise between run time, stability, and autonomy [2]. For that reason, it is essential to find a solution that considers the right choice of hardware implementing systems and software algorithm optimization to achieve the desired need for a well-defined application. Thus, it relies on hardware mechanisms to reduce consumption, but it uses the software to realize a dynamic adaptation of the consumption according to the load of the system. In practice, realtime motion estimation of objects remains a potential problem in most applications using these types of image processing algorithms. As a consequence, the system must present high stability, ease of implementation, and low run time.

Several methods have been used to estimate object movement in a real environment. Motion estimation is an important technique in many applications. Furthermore, it can be defined as the apparent velocity field observed between two successive images of the same scene [3]. This operation is known as the "optical flow". The optical flow is considered the most effective method to generate good results for motion estimation [4]. There are two major methods that highlight the computation of the optical flow to solve the motion estimation problem: matching techniques and differential methods [5,6].

The problem with the computation of the optical flow appeared after the research carried out by Horn and Schunck and Lucas and Kanade, which proposed two methods for optical flow estimation using global and local regulations [7]. As a result of their

work, several researchers put forward alternative methods to develop in order to reduce the execution time with a higher precision level to obtain better realtime results. Nevertheless, the optical flow estimation performance is affected by many factors. The effect of lighting variations and video quality degradation on optical flow estimation cannot be overlooked [8]. In addition, the majority of those applied methods have not met the realtime criteria if high precision is required [9]. Whatever the applied method, the optical flow calculation remains a very expensive operation in terms of its execution duration.

In order to overcome the existing problems of the traditional moving average background model, we propose an accurate motion estimation based on calculating the optical flow of a moving object using the Lucas–Kanade algorithm applied on a local treatment region in the latest Raspberry Pi 4 (8GB Ram). In addition, to improve algorithm accuracy, reduce the volume of stones, and improve the treatment results, our approach consisted of implementing iterative pyramidal refinements with some derivation blocks. Furthermore, to evaluate the precision of the optical flow estimation, we needed to compare the estimated optical flow field and the ground truth optical flow vector. For this reason, we tested our proposed method in the Middlebury dataset. Then, we applied it to synthetic reference images to compare our work with others. Our suggested method solved the problem of execution time with high precision and reduced the computational complexity of the optical flow calculation. Subsequently, our proposed model performed a realtime requirement.

We organize the rest of our paper as follows: In Section 2, we review the related work. In Section 3, we describe our proposal for optical flow estimation. In Section 4, we demonstrate the experimental results of our approach implemented into an embedded system. Finally, in Section 5, we present the conclusion to this study.

## 2. Related Work

### 2.1. The Principal Method for Optical Flow Calculation

Motion estimation is one of the very interesting problems in the image processing field. This problem consists of estimating from the temporal sequences of 2D images the apparent movement of the objects constituting a three-dimensional scene. As a result, the optical flow tries to find the vector field that connects two successive images in a video sequence [10,11]. Hence, we can deduce the global motion of the object by analyzing the direction of these vectors. Motion estimation is used in several applications such as video processing, video compression, vision, and object tracking.

Most optical flow estimation methods are based on a fundamental assumption: the light intensity is maintained between two successive images [12]:

$$I(x + w, t + 1) - I(x,t) = 0 \tag{1}$$

The conservation hypothesis can also be written in differential form [11,13]:

$$\frac{\partial I}{\partial t} = 0 => \frac{\partial I(x(t), y(t), t)}{\partial t} = \frac{\partial I}{\partial x} \times \frac{\partial x}{\partial t} + \frac{\partial I}{\partial y} \times \frac{\partial y}{\partial t} + \frac{\partial I}{\partial t} \tag{2}$$

$$\left[\frac{\partial I}{\partial x} \times \frac{\partial I}{\partial y}\right] \times \left| \begin{matrix} \frac{\partial x}{\partial t} \\ \frac{\partial y}{\partial t} \end{matrix} \right| + \frac{\partial I}{\partial t} = 0 \quad => (\nabla I)^t \omega + I = 0 \tag{3}$$

This formulation is the constraint equation of apparent motion. The assumptions made are the conservation of the luminous intensity, the conservation of the gradient of the intensity, global filtering, and a multiscale approach.

In what follows, two major methods highlight optical flow computation to solve the motion estimation problem [14]: the matching technique and the differential method.

### 2.1.1. Matching Technique

In this technique, the movement between two images is estimated by matching the elements of the first with those of the second [12]. The estimated motion can be represented

by a velocity field or indicative colors. According to this algorithm, the optical flow is locally constant; i.e., for each point, the displacement of nearby points under the optical flow is actually similar to that of the central point [13].

The structure of the algorithm is represented as follows. All points that present motion displacement in a frame are compared to those in the next image. A measure of the matching points between the two images is computed over a small region. This technique can be interpreted as matching small patches from the first image with small patches of the second one.

This method is local and simple. However, it presents intensive computation compared to the measurement stage of most differential methods. This problem is due to the number of comparison steps that depend on the displacement range. Its major advantage is that it is simple and parallelizable. Nevertheless, it becomes more complex for an important search space.

### 2.1.2. Differential Method

This method is based on spatio-temporal gradients of light intensity. It presents a good compromise between the search dimension and the object speed, but it remains a little more complex and difficult to implement in hardware.

It is possible to extract optical flow information through the assumptions of spatial and temporal image-brightness variations [13]. Suppose that $E = E(xl, x2, t)$, at location $i = (xI, x2)$, is the image brightness at time t on the image plane in a suitable system of orthogonal coordinates; the weakest possible assumption is the total temporal derivative of the image brightness.

### *2.2. Optical Flow Overview*

In the last decade, optical flow has become an alternative technique to estimate successive frames. In fact, optical flow is widely used in motion detection, robotics, visual navigation, and image processing [15,16]. Moreover, an important amount of research is based on optical flow estimation for vision systems.

Many approaches have been based on optical flow and suggested improvements in computer vision application using motion estimation. Ivan Vishniakou et al. [17] developed an optical flow estimation method for tracking treadmill ball motion in real time using a single high-resolution camera. Using calibration data, they determined the tracking accuracy and timing. The results demonstrated that the tracking system could be scaled up to be used on different size treadmill balls aside from the validity of the two-camera calibration procedure. The authors in [18] presented a realtime estimation algorithm that had a capability of 30 fps through the use of, for a pan/tilt camera, GPU-based approaches. Such an algorithm was based on the optical flow method. The results provide realtime optical flow estimation with high accuracy. Research was also carried out by Mahraz [19] who proposed a multiscale and variational approach for optical flow computation to modify the iterative implementation of the Lucas–Kanade method. They introduced the method that broke down an image into three parts: structures, textures, and noise. This approach offered good subpixel accuracy, but it did not meet the realtime requirement in terms of computation time.

Based on optical flow estimation, the authors in [7] suggested an effective motion detection method. They could extract the motion object's complete boundary by combining the optimized vertical and horizontal flow. This method presented high accuracy precision of optical flow calculation, but it needed three images to calculate the optical flow of one frame. Therefore, it occupied a large amount of memory space, so it affected the execution time and consumed more energy. The writers in [20] proposed, in this context, a global optical flow based approach for the estimation of multicomputer velocity using off-the-shelf onboard sensors, including a downward-looking monocular camera, an inertial measurement unit, and sonar facing downwards in GPS-denied environments. In the [8] paper, an illumination adjustment mechanism was introduced for retinal processing to

reduce the illumination variation. An edge refinement structure, based on weighted neighborhood filtering, was introduced into the optical flow estimation. Used on the Middlebury dataset, the experimental results showed that this approach was robust against illumination, and preserved motion detail accuracy. Zhang et al. [14] integrated an optical flow subnetwork to generate a novel approach based on spatio-temporal localization. Precisely, the designed network showed an accurate and efficient optical flow estimation by using many consecutive RGB frames instead of two adjacent frames in a deep network.

Following this trend, a motion estimation application based on the technique of optical flow estimation by the Lucas–Kanade method was designed in our work. Our method was performed on the local area of an image to achieve a realtime requirement. Based on the advantages of the optical flow, a realtime motion estimation algorithm was implemented on Raspberry Pi 4.

## 3. Approach Overview and Results

### 3.1. Approach Overview

From this primordial synthesis, we chose to work on mapping methods to calculate the optical flow of a moving object in a video sequence. Being simple, reliable, and parallelizable, our chosen method is best suited for implementation on an embedded system.

The next equation presents the constant field motion [20,21].

$$v = arg\ min_v \sum_{p \in \Omega} \left( \nabla I(m, t) \times v + I_t(m, t) \right)^2 \tag{4}$$

The Lucas–Kanade method is based on the computation of temporal and spatial derivatives of an image [22]. To calculate them numerically, numerous solutions have been proposed using convolution masks, such as Sobel or Prewitt. However, the most suitable mask for variational methods is the one suggested by Horn and Schunck, which encapsulates the pixel in space and time.

The method of Lucas and Kanade is generally summarized as follows:

First, a moving object is specified, which is saved as a model reference. This image reference will be the object of a partial derivation with the global image.

$$E(u, v) = \sum (I(x + u, v + v) - T(x, y))^2 \tag{5}$$

This equation cannot adapt to the hypothesis of conservation of the constant flow over a wide period. Therefore, the equation of Lucas and Kanade is generalized to other 2D parametric motion models by introducing a "warp": function W [22].

$$\sum (I(W([x, y]; P)) - T([x, y])))^2 \tag{6}$$

This approach implicitly assumes that the errors in the image data have a Gaussian distribution with a zero mean. Then, until the window reaches a certain percentage of value outliers, statistical analysis may be used to detect this noise and reduce its weight accordingly. The Lucas–Kanade method can be applied just when the object displacement between the two frames is small enough for the differential equation of the optical flow to hold. This is often less than pixel spacing [23]. Subsequently, when the flow vector exceeds such a certain limit, for example, in warped document registration or stereo matching, it is possible for the Lucas–Kanade method to be still utilized to refine some coarse estimate of that image. For example, this is performed using other means such as extrapolating the flow vectors computed for previous frames or running the Lucas–Kanade algorithm on reduced-scale versions of the images.

### 3.2. Improvement

To improve algorithm accuracy, we applied two principles that express the same idea at two different scales (Figure 1):

- Iterative refinement: The main idea of this technique was to execute the algorithm n times related to the previous results. Therefore, it enabled a reduction in the direct error at each iteration. However, the run time rose depending on the iteration number. Then, the method converged when the algorithm was stable [23].
- Pyramidal implementation: the pyramid was produced by the resampling of successive frames. A pyramid was defined with various levels (Ln), typically in the range of two to four, as depicted in Figure 1. At each level of the pyramid, the image was subsampled by a factor of two for the two successive images considered. While level zero corresponded to the initial image, level Ln corresponded to the coarsest level [24]. At the Ln level, the optical flow was computed, which was then propagated to the lower level by translating image "f1" a priori calculated at a higher level. The algorithm was repeated to reach level zero, which corresponded to the initial image. Then, the final optical flow was recuperated [25]. This multiresolution approach reduced the volume of stones and improved the results of treatments.

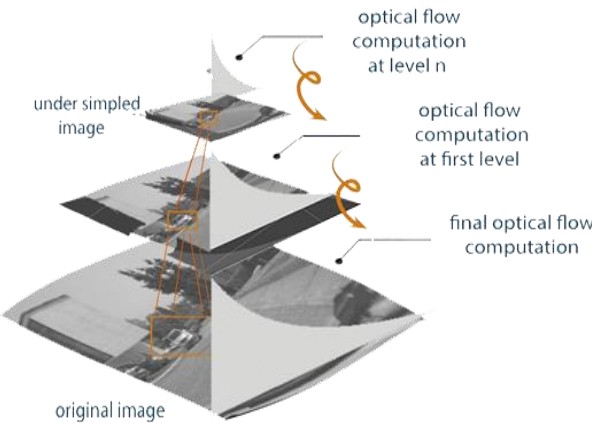

**Figure 1.** Iterative pyramidal refinement.

### 3.3. Algorithm

Taking into account the previous, we chose the optical flow estimation algorithm with the iterative pyramidal implementation of the Lucas–Kanade method and introduced regularized least squares. The precise description of the implementation of our algorithm is given by the following algorithm (Algorithm 1):

---

**Algorithm 1** Pseudo code for min algorithm of pyramidal refinement

---

**input**: frame 1, frame 2, level number, patch size
pretreatment and filtering
pyramid building
**for** L = 0 to L = pyramid level **do**
    spatial and temporal derivation
    velocity interpolation
        **for** i = 0 **to** i = iteration number **do**
optical flow refinement
computing final optical flow
**output**: final optical flow

---

From two successive images, we created a Gaussian pyramid for each image. Then, the algorithm began with interpolation of the flow field of the upper level, and after, it was applied for each pixel. We noticed that the code described was similar to the one to calculate the derivatives on each patch. This latest version avoided redundant calculation and side effects. In addition, the moved index was calculated for the whole moment, which gave low accuracy because it lost the subpixel by calculating the round of the field to

the previous level. This problem could be solved by performing the interpolation of the derived images as described below.

Additionally, we adjusted the End Point Error (EPE) averages on full-resolution data by factor two. This made the evaluation fairer and easier, as the EPE would be significant as it was measured in pixels, at a higher resolution. Moreover, the average remained stable with the variation of the data.

### 3.4. Optimization

Video processing requires a large memory space that subsequently affects the run time level. When the entire image is processed, a slow rate at the video display level and a delay in the motion detection operation are observed. Therefore, a technique that ensures the performance and memory gain was used. Its principle was to perform global and local treatments at the same time. The programed algorithm is presented in Figure 2.

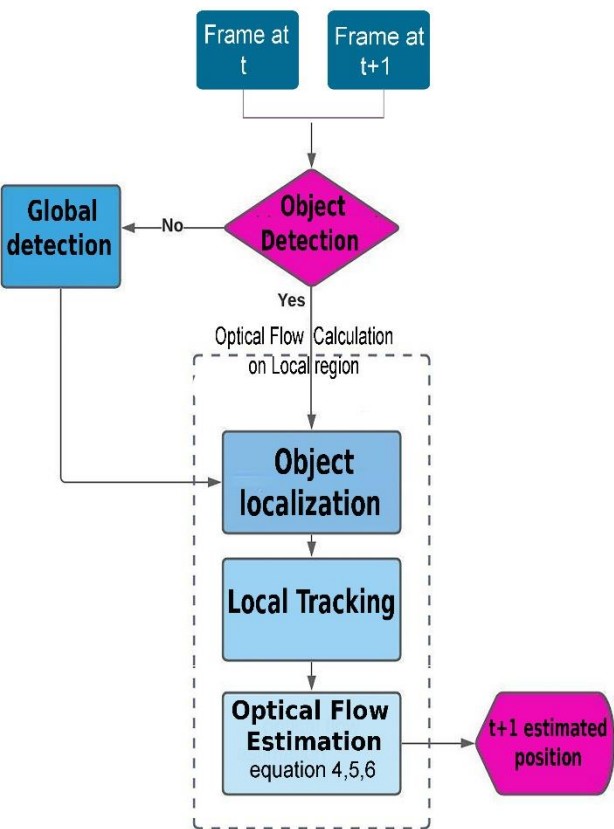

**Figure 2.** Algorithm diagram of the proposed method.

In fact, at the beginning, the program conducted a global treatment to find the desired object on the entire image. Subsequently, to localize the detected object, the barycenter of its edge contours was searched. The detected object was characterized by its color, edge shape, and localization points.

Then, the local treatment was activated to make motion detection at a small adjustable section. This portion of the image was located at the coordinates found during global processing. Therefore, the displacement of the object affected the position of the local processing area following the object's movement.

This method provided good results at the execution time and the memory allocation level. However, when the object moved at a higher speed, it may not have been detected in this small area resulting in being lost.

The solution was to conduct a global treatment another time to detect its position. This implied that the run time of the algorithm depended on the speed of object movement.

When the speed of the object increased, the number of the global treatment process rose, hence affecting the efficiency of the algorithm.

It was also noticed that this program was effective with a maximal movement speed of the object, thus allowing it to be tracked by the local area. It was inferred as well that if the speed of the object exceeded the speed maximum, the global processing would be activated automatically. Therefore, the effectiveness of the program with the global treatment activations' numbers can be evaluated. To reduce the number of global treatment activations, the motion estimation technique (Lukas and Kanade) described earlier was used.

When the object started moving it could be detected and located, and it could be identified by its coordinates and some other characteristics such as its shape and color. The motion estimation between two successive images was to match the elements of the first with those of the second, which could be linked by the motion estimation vector (or color indication). This estimated vector presented the position of the object at "t" and estimated its location at "t + 1".

It was also stated that the local processing area followed the object displacement. The motion estimation technique was applied to the algorithm. Then, the local processing zone moved into the estimation position. As a result, when the object was in the "t" position, the local treatment zone became the "t + 1" position. Consequently, this operation may have reduced the number of global treatment activations.

To see the efficiency of the algorithm, it was necessary to compare the movement estimated by the real movement each time. A correction operation could be introduced to improve the treatment (Figure 3).

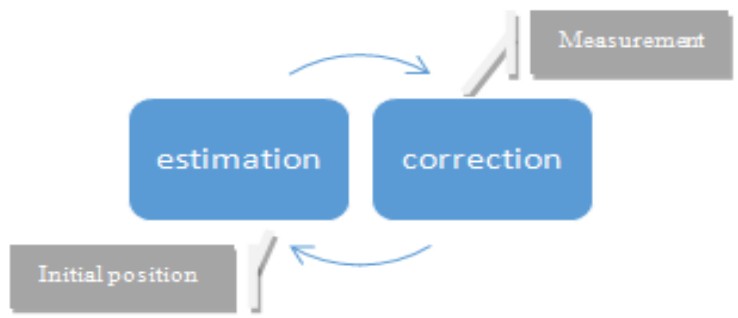

**Figure 3.** Correction process.

In the next section, the estimation algorithm's combination with the local treatment process is discussed and the simulation results are presented. The color map is used to present the estimated optical flow.

The experimental results showed that there was a marked improvement in the quality of the estimate and even in terms of execution time.

## 4. Result and Discussion

### 4.1. Evaluation

To evaluate the precision of optical flow estimation, we needed to compare the estimated optical flow field and the ground truth optical flow vector. For this reason, we tested our proposed method on the Middlebury dataset (Figure 4).

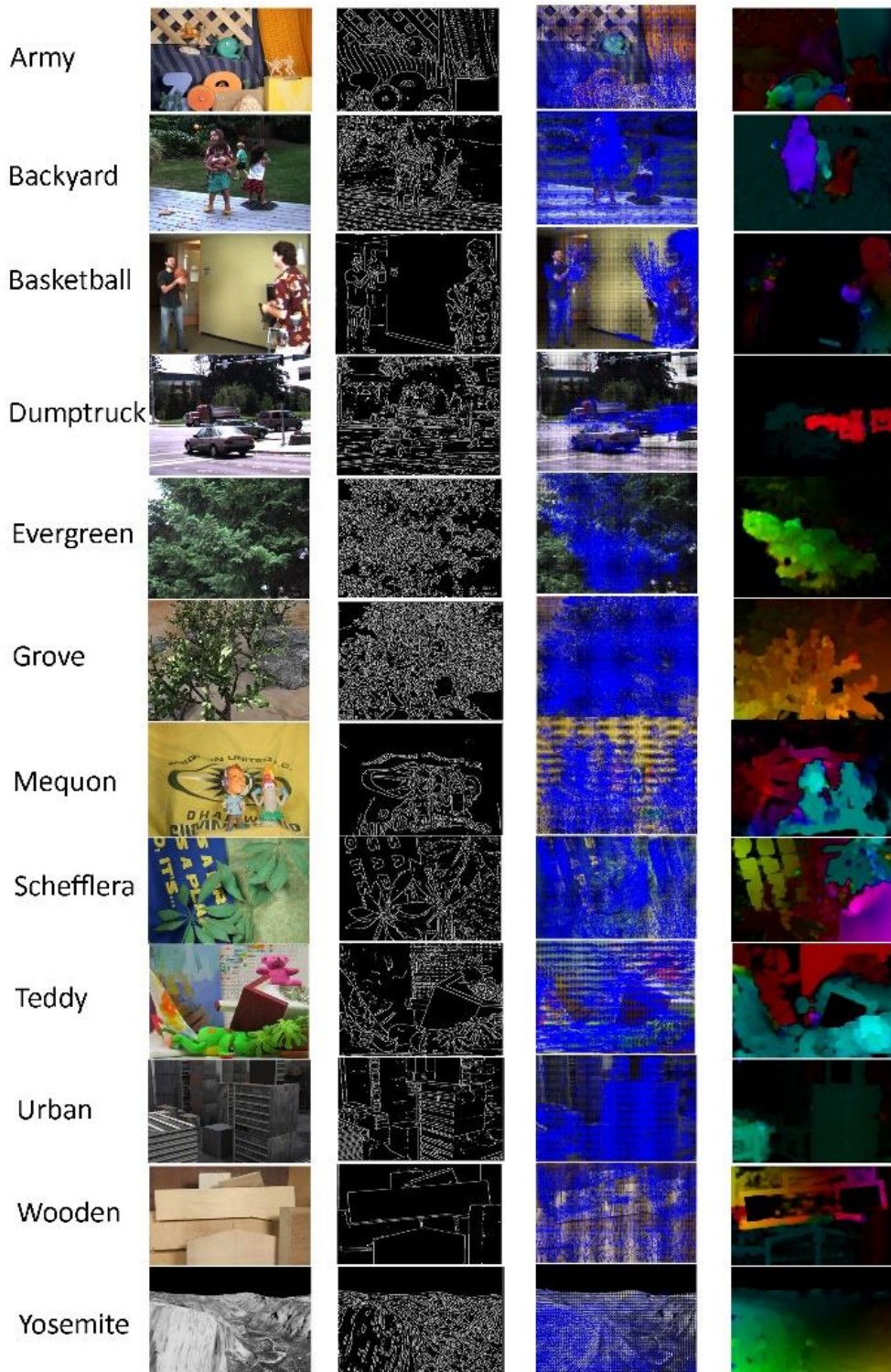

**Figure 4.** Results of optical flow estimation tested in the Middlebury Dataset.

Then, we applied it to synthetic reference images to compare our work with others. One of the most used sequences is the one called "Yosemite", which was named after the American park containing the valley presented in Figure 5. The advantage of this sequence is that it contains complex movement (rotations and translations).

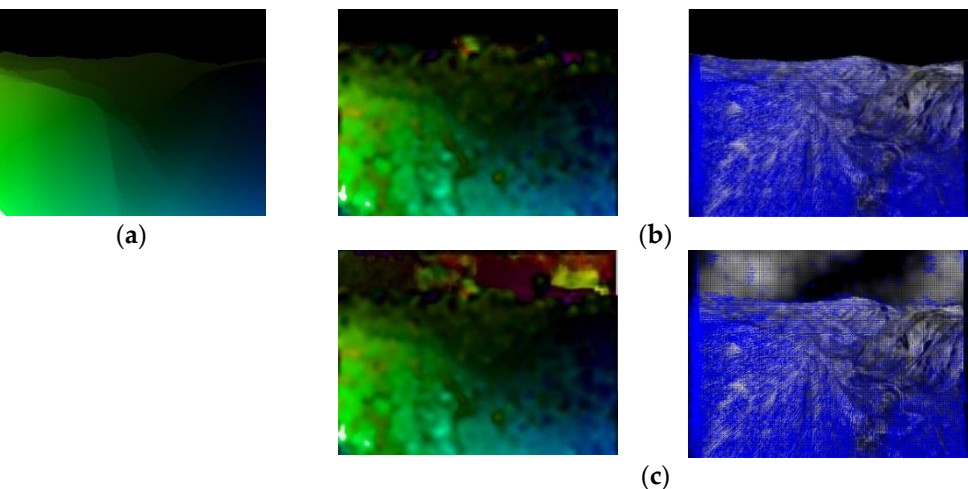

**Figure 5.** Real flow and estimated flow on synthetic images of Yosemite. (**a**) Real flow. (**b**) Our approach for optical flow estimation on synthetic images of Yosemite without clouds. (**c**) Our approach for optical flow estimation on synthetic images of Yosemite with clouds.

Therefore, from the real flow, we can, on the one hand, evaluate the performance of the developed algorithm. On the other hand, we can study the influence of the parameters on the error, to choose them as judiciously as possible.

*4.2. Error Measurement*

There were several methods used. The Angular Error (AE) and the EPE were the most used ones. These are, respectively, measured as follows:

The AE is defined as the angle between the estimated optical flow field and the groundtruth optical flow vector:

$$\text{AE} = \frac{1}{|\Omega|} \sum_{\Omega} arccos \left( \frac{u_r u_c + v_r v_c + 1}{\sqrt{(u_r^2 + v_r^2 + 1)} \sqrt{(u_c^2 + v_c^2 + 1)}} \right) (E1) \tag{7}$$

The EPE is defined as the distance between the endpoint of the estimated and the groundtruth flow field:

$$\text{EPE} = \sqrt{(u0 - u1)^2 + (v0 - v1)^2} \tag{8}$$

Figure 6 illustrates the gap between the two vectors.

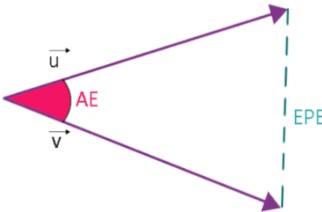

**Figure 6.** Difference between AE and EPE.

Our method had three parameters to edit: the refinement iteration number per level, the number of levels, and the patch size where the velocity was considered constant.

We set the iterations number as one, and we varied the patch size. For the "Yosemite" images, three was the maximum number of levels allowed, because the initial image was $640 \times 480$ pixels. After that, we kept it at three. The results are depicted in Figure 7.

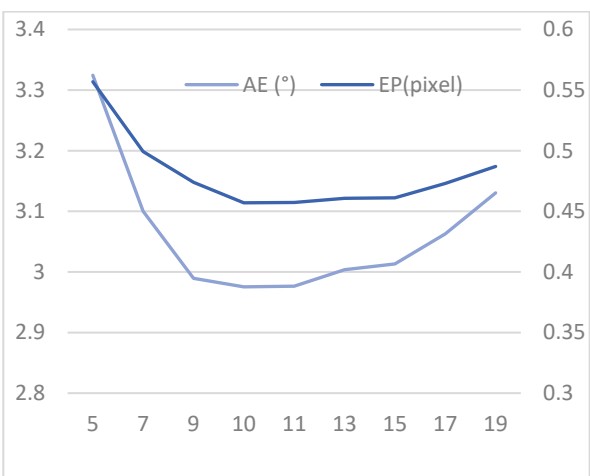

**Figure 7.** Error related to patch size.

The best patch sizes presenting minimal errors, in this sequence, were $11 \times 11$, $10 \times 10$, and $9 \times 9$. Then, we chose the $10 \times 10$ patch size as it presented the best result with varying the number of iterations.

Figure 8 displays the results of implementation.

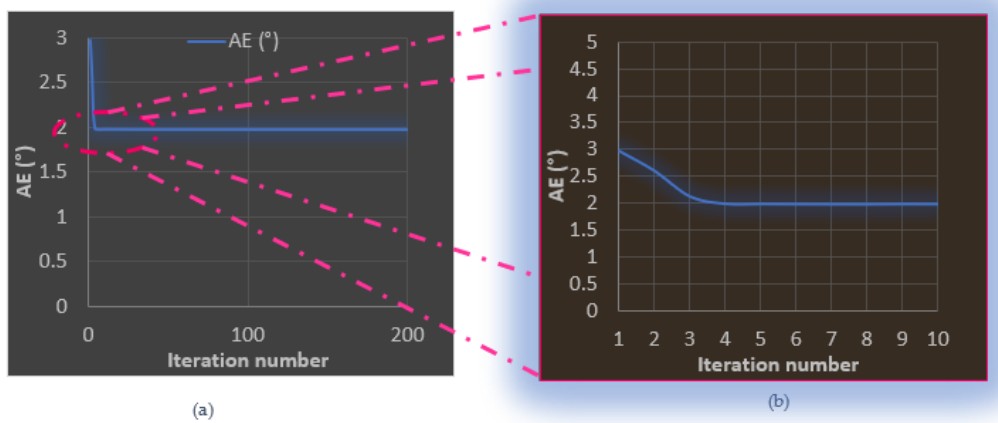

**Figure 8.** AE related to iteration number.

In Figure 8a, we note that the decrease in AE depended on the iterations number. Nevertheless, the result remained stable at a certain level. Therefore, to see the stability zone beginning, we focused in on the diagram.

As we have stated, the run time increased with the iteration number. From the results shown in Figure 8b, the best compromise in terms of run time and accuracy was three iterations. Therefore, we selected it, because a larger number of iterations did not improve the accuracy of the system.

*4.3. Comparison*

In this table, the comparison of the average AE of our method with three others of [26–28] is given. The performance of optical flow estimation of these algorithms was given using the Middlebury dataset image sequences. The comparison was achieved by calculating the average AE between the estimated optical flow field and the groundtruth optical flow vector. Table 1 shows the average AE retained by applying the optical flow estimation methods on the Yosemite sequence with clouds and without clouds.

**Table 1.** Comparison of the average AE with other methods.

| Image Sequences | Methods | | | |
| --- | --- | --- | --- | --- |
| | Ibarra's [26] | Chang's [27] | Zhenghua [28] | Proposed |
| Yosemite without clouds | 0.77 | 0.35 | 0.25 | 0.19 |
| Yosemite with clouds | 0.76 | 0.36 | 0.23 | 0.21 |

The results obtained by [27,28] were better than those produced by [26] but the obtained optical flow seemed sparse. Therefore, they were missing many useful details especially local moving information. According to [28], the method of [27] in visual comparison shows that it could generate a higher robustness than [26]. The largest AE of each index shown given by [26] was an expected result of the measurement of the dense optical flow with low robustness. Therefore, we can say that the estimated optical flow used in [27,28] was almost close and presented good accuracy.

Similarly, compared to the other three methods, our suggested method showed the least error for each sequence which can be seen in Table 1, proving the effectiveness of our proposed method.

As illustrated in this table, our method outperformed [26,28]. Indeed, the authors in [28], to calculate the optical flow, combined the horizontal and the vertical flow, which could not be exposed by the direct position of the vertical flow and horizontal flow. Therefore, the object could not be fully detected. In view of this, this method presented a lower accuracy relative to our proposed method.

In addition, the visual observation of the obtained result, especially marked with color code field representation, presented a clear optical flow and performed a best estimated local optical flow, which could be useful for the analysis image content details. Consequently, our method provided a more accurate optical flow estimation with respect to the groundtruth optical flow and kept more useful details of the image.

From the analysis of error results, we concluded that the best parameters assigned to our approach for the optical flow calculation were: a $10 \times 10$ patch size, a three-level pyramid, and three iterations. Therefore, we kept it to estimate, in the next part, the optical flow motion in a video sequence.

*4.4. Computational Coast*

As we have specified, our approach varied between global and local treatments. In this part, we extracted the results of the calculation of the optical flow on a real scene whose target object was a moving vehicle.

In this section, the implementation results of our algorithm, conducted by Raspberry Pi 4 with a 64-bit quad-core Cortex-A72 processor and 8GB of RAM, are provided, as depicted in Figure 9.

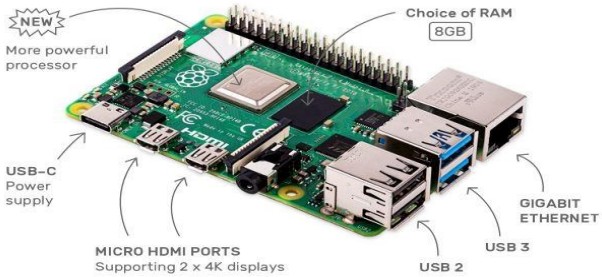

**Figure 9.** Raspberry Pi 4 8GB of RAM.

With the most recent Raspberry, so far, the Pi 4 8GB version is the most powerful of the Pi 4s and retains exceptional capacities compared to the Pi 3 and Pi 3 model B +. In

addition, it offers an increase in the processor speed, the multimedia performance, the memory, and the connectivity over the previous generation. Moreover, it can support more memory while maintaining backward compatibility and similar power consumption.

The program code was performed by Qt Creator tools with C++ language in the Rasbian operating system with OpenCV 3.0.1. The experimental result was obtained by executing the program with the "Profiling" function execution into the Qt creator.

To compare the different estimates obtained, it was necessary to use the universal visual representation of the velocity field. Many solutions were possible to represent the calculated optical flow. The most commonly used was the trace of the field velocity vectors superimposed on the initial image. The disadvantage was that we did not visualize the dense field. Thus, the detected points may have been masked. Therefore, we calculated an incorrect interpretation.

We could also represent the gray levels respectively by u and v. The most interesting solution was the use of a color map, which enabled representation of the direction of the flow as well as its intensity in a dense way. The color map is shown in Figure 10.

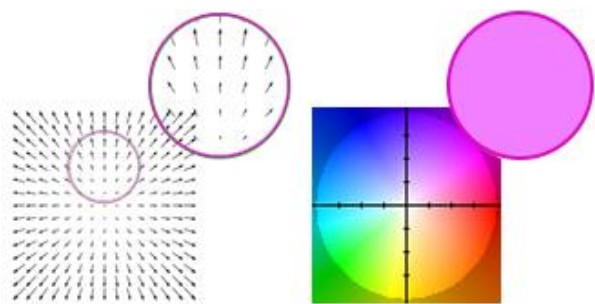

**Figure 10.** Color map.

The velocity vectors are represented by the colors contained in this circle. Each velocity vector corresponds to a defined color indicated in the circle. This type of representation allows a dense representation and makes it possible to visualize very quickly the coherence of the results, the sensitivity to noise of the method tested, and the aspect of the flow.

The execution of our algorithm with three pyramid levels, a patch size of $10 \times 10$ pixels, and three refinement iterations for all frames is shown in Figure 11.

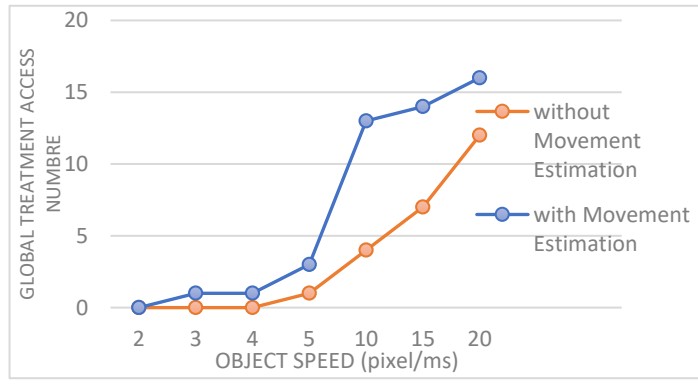

**Figure 11.** Frequency of global treatment access.

In a first test on the efficiency of the algorithm, we applied our approach to a video sequence containing a vehicle moving with different speeds, as described in Figure 12. The video sequence contained 300 frames.

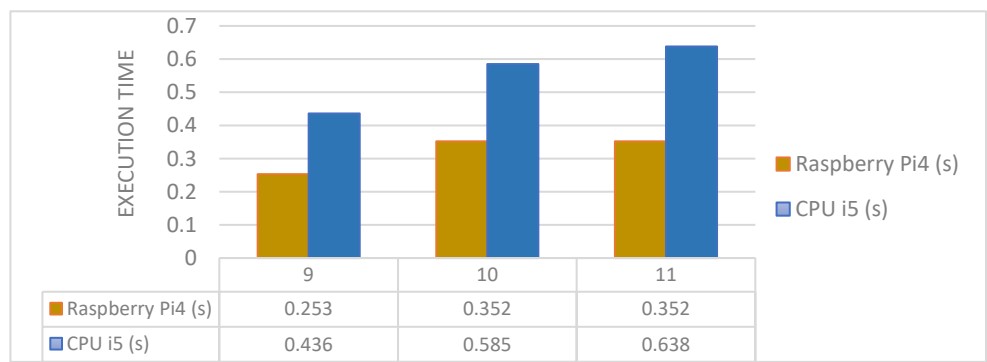

**Figure 12.** Run time performance of Raspberry Pi 4 compared to CPU i5 according to patch size.

As a consequence, we saw how much access to global processing was in this sequence. We recall that the algorithm, in the first place, processed the entire image to locate the vehicle. Then, it calculated the optical flow in its neighborhood (region of $50 \times 50$). If this object left the local processing region, the program conducted the global processing to find the object in the whole image. We recall that the global processing was more expensive at the level of execution time and memory allocation.

Therefore, we compared our algorithm that used the motion estimation of the local processing region with an algorithm without any motion estimation.

As has been mentioned, the overall processing of the whole image was very expensive in terms of execution time and memory allocation. Thus, our goal was to reduce the frequency of access to this treatment. The results clearly showed that our method, based on the estimation of vehicle movement, was very efficient in reducing access to the overall image processing, which clearly showed its positive effect on reaching the realtime criterion.

*4.5. Run Time*

In the previous test, we deduced that the best estimation was performed between the $9 \times 9$ and $11 \times 11$ patch size. To express the performance of this work, we provided a comparative analysis of the run time for the vehicle tracking algorithm implementation, in a real video sequence, between Raspberry Pi 4 (8GB ram) and CPU CORE i5, by testing the program with different patch sizes (Figure 12).

The results were completely encouraging. Moreover, the Raspberry execution time observed on the $640 \times 480$ sequence for four levels, three iterations, and different patch sizes was more efficient than the CPU one. Hence, the realtime challenge was achieved, after going through a methodical phase code optimization.

Our approach was tested on a video with a fixed camera. In this case, the background was fixed and the targeted objects were mobile. In this environment, we achieved a very good compromise between the precision of the motion estimation and the computation cost. Nevertheless, with a mobile camera, the scene would be different, because all the components of the image would be mobile, which generates other problems of identification and estimation of the target object motion. In addition, our approach was treated sequentially. Therefore, parallel implantation was particularly suitable. This opened up many perspectives for the development of real applications and could be a good way to accelerate the method even further.

From the results obtained, shown in Figure 13, we were able to locate the moving objects. In addition, referring to the color map of Figure 10, it was possible to identify the estimated direction of vehicular movement. The speed fields were not scattered. This was due to the elimination of unnecessary information. Then the estimated flow was not disturbed. Therefore, we could produce a high precision optical flow estimate.

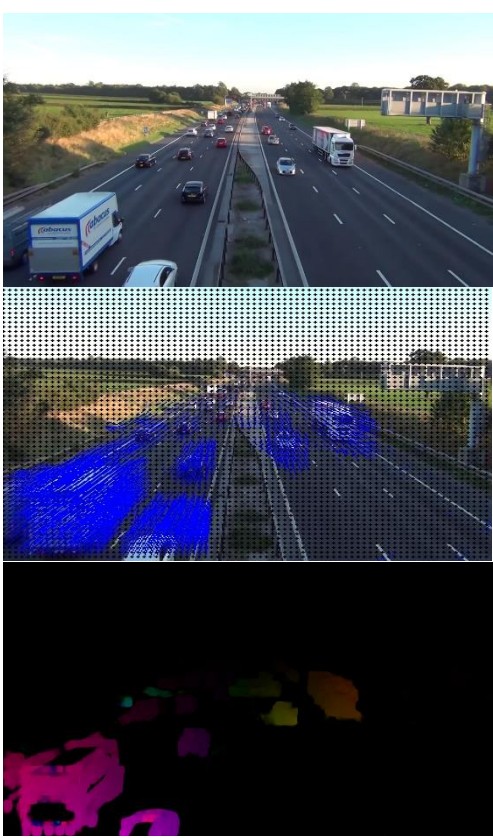

**Figure 13.** Estimated optical flow on real road traffic video.

## 5. Conclusions

In the context of improving an object tracking system, we have proposed a technique that enables quantification of the movement of a mobile object. This technique is based on the calculation of optical flow. Its principle is to proceed with the localization of the target object. Then, the Lucas–Kanade method of motion estimation on a local area is carried out.

In this paper, we have developed an accurate motion estimation based on the Lucas–Kanade technique implemented on the latest version of Raspberry Pi 4. To improve algorithm accuracy, reduce the volume of stones, and improve the treatment results, we applied the pyramidal iterative refinement. To evaluate the precision of the optical flow estimation, we compared the groundtruth optical flow field and the estimated optical flow vector. We tested our proposed method in the Middlebury dataset. Then, we applied it to synthetic reference images to compare our work with others. The evaluation section proves that our proposed method has less EPE and the angular errors attain competitive results with an average as low as 0.2. In addition, while at the same time performing high accuracy and satisfying the realtime constraint, the experimental results showed that our improved algorithm reduces the run time, in a significant way. We note that, specifically in the pyramid building process, our proposed algorithm was highly parallelizable. Then, parallel implementation could produce high performance in our future work.

**Author Contributions:** Methodology and software, A.A.; writing—review and editing, H.B.F. and A.A.; supervision and project administration, C.S. All authors have read and agreed to the published version of the manuscript.

**Funding:** This research received no external funding.

**Conflicts of Interest:** The authors declare no conflict of interest.

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
