# Peer review of "Accurate Realtime Motion Estimation Using Optical Flow on an Embedded System"

_electronics, doi:10.3390/electronics10172164_

Round 1

Reviewer 1 Report

Dear authors, while the presentation is nice in shape, there are few comments and/or suggestions to strongly improve the manuscript. While there is some interest in this type of research, Please reconsider the paper as:

  1. The significance of the study is not clear to me and there is a serious literature review gap in this paper. The literature state-of-the-art contains too few references. The related work section must be extended with 15-20 additional relevant sources (IEEE, Springer, Elsevier, MDPI). Also, the cited references are inappropriate and must used in accordance wih the template. (please reconsider [1], [2], [3] ..... in the paper text etc.)
  2. Clarify better the advantages of this paper in the introduction section because in the literature a lot of recent papers consider the same proposed approach.
  3. The method itself is not well described. There are a number of (simple) mathematical quantities that are defined, but their use in the proposed planning model is not precisely defined.
  4. Moreover, the authors must present the limitation of the considered method.
  5. The results are abundant but the results' analysis is not enough. However, I cannot see deeply analysis related to them and cannot understand the meaning of results. Please add more analysis.
  6. The conclusion looks like a summary. It would be better to rewrite it and add many effective results in the Conclusion section.
  7. Another essential improvements. I strongly consider that the distinguished authors did not pay attention to the paper text. Please modify or change the following aspects:

- Line 12 – Abstract... must be erased.

- Lines 41-42 – the references between [2] and [12] are missing. Please reconsider in accordance with the paper template.

- Line 89 – figure 2... Please see that the first figure is presented on page 5 at line 200, and figure 2 at 238. Please reconsider these aspect.

- line 228 – EPE, probable means End Point Error (EPE). Please use as an abbreviation.

-line 341 – Please reconsider the subtitle!!!

The authors must strongly consider the aforementioned suggestions and resubmit.

Reviewer 2 Report

This article implements a local treatment region by Raspberry Pi 4. The main problem and motivation of this topic are unclear. The proposed embedded system is also unclear. The authors should provide the system architecture of the proposed system and demonstrate their prototype of the proposed system. The texts in figure 2 are not clear. The authors should improve the quality of the figures. The values of Table1 should be given their units. The right side of figure 11 should be removed. Execution time of figure 11 should be given a unit.

Misc. 

Article title:  "accurate" should be "Accurate"

Line 12: One more "Abstract:" should be removed.

Round 2

Reviewer 1 Report

The authors have satisfactorily addressed most of my comments. Moreover, I have some minor suggestions:

  1. Please renumber the cited references in the paper text, because the paper starts with [1].
  2. The multiple citations are not accordingly for the Journal ethics. Please reconsider them [29, 30, 31, 32, 33] with a short explanation. If they consider the same aspect, please use only one or two.
  3. The limitation of the proposed approach is not yet emphasized.

Reviewer 2 Report

The revised paper has been improved the quality. Hence, I think that the revised paper can be considered to accept for publication. Figure 2 should be scaled.
